# Clinical and Clinicopathological Features of 108 Dogs Infected with *Babesia gibsoni* in Hong Kong

**DOI:** 10.3390/ani15050645

**Published:** 2025-02-23

**Authors:** Karen Chan, Paweł M. Bęczkowski, Angel Almendros

**Affiliations:** 1Department of Veterinary Clinical Sciences, Jockey Club College of Veterinary Medicine and Life Sciences, City University of Hong Kong, Kowloon Tong, Hong Kong SAR, China; 2CityU Veterinary Medical Center, City University of Hong Kong, Kowloon, Hong Kong SAR, China

**Keywords:** *Babesia gibsoni*, canine babesiosis, dog, arthropod-borne disease, clinicopathological features, Hong Kong

## Abstract

*Babesia gibsoni* is a tick-borne pathogen which infects canine red blood cells and can cause significant health issues in dogs. In this study, we analyzed medical records of infected dogs to better understand the clinical signs and identify patterns in blood tests and urinalysis that can aid in the diagnosis. The most common clinical signs included anorexia, pallor, lethargy, pigmenturia, and fever. Most dogs showed anemia and thrombocytopenia. Other common findings included changes in liver enzymes, hyperproteinemia, and abnormalities in urine, such as proteinuria and bilirubinuria. These findings can aid veterinarians in increasing their suspicion of babesiosis, prompting the use of molecular diagnostics, detecting the disease earlier, and improving the treatment outcome of affected dogs.

## 1. Introduction

The hemoprotozoan parasite, *Babesia gibsoni*, is a small pleomorphic piroplasm identified inside red blood cells. *B. gibsoni* belongs to the Piroplasmida order, Sporozoasida class, and Apicomplexa phylum [1]. The piroplasm ranges from 1 to 3 μm in size and varies in morphology from oval to ring or comma-shaped [1]. Similar to other *Babesia* spp., it is primarily transmitted by ticks, namely *Haemaphysalis bispinosa*, *Haemaphysalis longicornis*, and *Rhipicephalus sanguineus*, belonging to the Ixodidae (hard tick) family [2]. However, non-vectorial routes of transmission, including transplacental and direct transmission through blood via fights, are also possible [3,4,5,6]. Being a geographically widespread disease, it has been detected in Asia, the Middle East, Africa, Europe, and some parts of the United States [7,8,9]. In Hong Kong, it has been reported that babesiosis is the most common arthropod-borne infection in dogs, with *B. gibsoni* (Asian genotype) accounting for most cases, with a reported prevalence of 3.7% and up to 27% in pets with suspected tick fever [10,11,12]. Risk factors for infection include mixed breed dogs that spend time outdoors and are therefore exposed to ticks and are generally younger than 10 years old [12]. Horizontal transmission through bites, however, is reported in pure-breed dogs like American Pit Bull Terriers, Staffordshire Terriers, and Tosa Inus [6,13].

The presentations of the infections can range from hyperacute and acute to chronic, and can manifest as asymptomatic as well as complicated symptomatic cases. The hyperacute form of the disease is rare and occurs mostly in young puppies [14]. Shock and extensive tissue damage are rare and have been reported with hyperacute onsets in fighting dogs [6]. The acute form of the disease is commonly presented with non-specific clinical signs, including pallor, fever, anorexia, weakness, and splenomegaly [15]. Hemolytic anemia and thrombocytopenia are the most common findings in infected dogs [16,17]. Other documented clinicopathological alterations include non-specific changes in the leukogram, elevation in liver enzymes, hypoalbuminemia, hyperglobulinemia, and azotemia [16]. The chronic form is often subclinical but might present with intermittent fever, lethargy and weight loss [18], resulting in dogs remaining carriers and reservoirs of *B. gibsoni*, leading to recrudescence of signs under stress or immunosuppression [2]. Complicated forms are more commonly reported with the more virulent *B. rossi* or *B. canis* infections [19,20,21]. However, clinical manifestations extending beyond the effects of hemolysis have also been reported anecdotally or in case reports in dogs infected with *B. gibsoni* [22,23,24,25]. As such, bicavitary effusion, disseminated intravascular coagulation (DIC), glomerulonephritis, renal failure, pulmonary edema, and acute respiratory distress syndrome have been reported as complications [22,23,24,25]. The most reliable diagnostic method for detecting *B. gibsoni* is the polymerase chain reaction (PCR) in peripheral blood samples, due to its high sensitivity and specificity despite a low level of parasitemia [26]. Low parasite load in chronic stages or artifacts make light microscopy less sensitive when compared to PCR. However, as PCR is not often readily available in-house and requires time to process through external facilities, treatment is often initiated based on presumptive diagnosis derived from history, clinical presentations and clinicopathological data while results are pending.

Published studies on *B. gibsoni* infection in dogs in Hong Kong have mainly focused on investigating treatment, prevalence and risk factors [10,11,12,27]. However, clinical presentations and clinicopathological findings, such as alterations in hematology, coagulation time, biochemistry, and urinalysis, have not been well elucidated and are less defined. A comprehensive and detailed analysis helps identifying the likely changes to be seen in infected dogs, thereby facilitating a more efficient diagnosis of the infection. This study, therefore, aims to further describe the clinical signs of dogs infected with *B. gibsoni*, and to determine the pattern of alterations and magnitude of changes observed in clinicopathological data. It is anticipated that such analysis might offer clinicians insights into when to suspect the presence of *B. gibsoni* infection, so it can be included in their differential diagnoses. This would prompt the earlier use of PCR and lead to a timely intervention for a better treatment outcome.

## 2. Materials and Methods

### 2.1. Study Design and Patient Selection

A retrospective analysis was conducted on dogs that presented to an emergency and specialty veterinary hospital in Hong Kong between 2012 and 2022. Data were obtained from 264 dogs that had a positive PCR diagnosis. As well as a positive PCR, the inclusion criteria for the study was for dogs to have a complete blood count (CBC) and serum biochemistry results including blood urea nitrogen (BUN), creatinine (CREA), total protein (TP), albumin (ALB), globulin (GLB), alanine aminotransferase (ALT), alkaline phosphatase (ALP), γ-glutamyl transpeptidase (GGT), and total bilirubin (TBIL). Dogs with incomplete data, dogs with pre-existing co-morbidities, and dogs younger than 1 year old or older than 9 years old were excluded from the study to minimize the associated interference of these factors on the results. A total of 108 dogs that met the inclusion criteria were therefore selected for the study.

### 2.2. Data Collection

Medical records were obtained and analyzed using RxWorks Veterinary Software (Covetrus, Inc., Tauranga, New Zealand). Age, sex, breed, presenting signs, physical examination findings, blood smear evaluations, and saline agglutination results were reviewed and recorded. Numerical results for CBC, serum biochemistry, and coagulation times, along with whether they were decreased, within the normal interval, or increased, were generated from in-house automated blood analyzers, ProCyte Dx Hematology Analyzer, Catalyst One Chemistry Analyzer, and Coag Dx Analyzer (IDEXX Laboratories, Inc., Westbrook, Maine, USA), respectively. Canine pancreatic lipase immunoreactivity (cPLI) results obtained by SNAP cPLI Test (IDEXX Laboratories, Inc., Westbrook, Maine, USA) were also analyzed. Urinalysis results were determined using IDEXX UA Strips analyzed with the VetLab UA Analyzer (IDEXX Laboratories, Inc., Westbrook, Maine, USA). Molecular diagnosis was conducted at the Veterinary Diagnostic Laboratory (VDL) of City University of Hong Kong or at the Department of Microbiology, Veterinary Division, Hong Kong University (HKU) using anticoagulated blood. A tick fever panel tested for *Babesia* spp., *B. gibsoni* and *E. canis*. An ELISA test (SNAP 4Dx Plus, IDEXX Laboratories, Westbrook, ME, USA) was used to test for other vector-borne diseases in some of the dogs. All the collected data represented the findings recorded by the attending clinician at presentation and before a diagnosis or specific treatment had been established.

### 2.3. Statistical Analysis

The results were analyzed using SPSS for Windows (version 29.0.2.0, SPSS Inc., Armonk, NY, USA). The data were tested for normality by skewness. The median was calculated and indicated when a not normal distribution was present. Continuous variables are reported as the mean with their respective standard deviations, as well as maximum and minimum values. Categorical variables are expressed in frequency and percentages.

## 3. Results

The mean age of the 108 dogs was 5.63 ± 2.37 years, with the most common age being 8 years old (28/108, 25.9%). A total of 38.0% (41/108) and 62.0% (67/108) were female and male, respectively. Among the total population, 46.3% (50/108) were male neutered, 33.3% (36/108) were female spayed, 15.7% (17/108) were male entire, and 4.6% (5/108) were female entire. There were 29 breeds documented, of which the most represented breeds included mongrel dogs (26/108, 24.1%) and Poodle (20/108, 18.5%).

The presenting signs and physical examination findings are shown in Table 1. The most common clinical sign was anorexia (73/108, 67.6%), followed by pallor (57/108, 52.7%), lethargy (49/108, 45.4%), and pigmenturia (38/108, 35.2%). Tick exposure was reported in only a few dogs (5/108, 4.6%). These initial findings were recorded as the cause for hematology, biochemistry, and subsequent serological or molecular testing.

Blood smear evaluation and saline agglutination were conducted in 58/108 (53.7%) and 19/108 (17.6%) cases, respectively. Out of the 58 blood smear results, spherocytosis and polychromasia were observed in 46.6% (27/58) and 43.1% (25/58) of the cases, respectively. Anisocytosis was observed in 37.9% (22/58) of the cases, and an equivalent proportion, 37.9% (22/59), presented with intraerythrocytic piroplasms. Among 19 cases with saline agglutination results, 15.8% (3/19) and 84.2% (16/19) had positive and negative results, respectively.

Severity of anemia was classified based on HCT as severe (HCT < 20%), moderate (20% ≤ HCT < 30%), and mild (30% ≤ HCT < 37.3%) [28]. We established the severity of thrombocytopenia based on deviation from normal parameters [29], and it was categorized as severe (PLT < 30 K/μL), moderate (30 K/μL ≤ PLT ≤ 100 K/μL), and mild (100 K/μL < PLT < 148 K/μL). Table 2 shows a summary of descriptive statistics for CBC parameters. The most observed hematological alteration was thrombocytopenia (98/108, 90.7%), with 11.2% (11/98), 53.1% (52/98), and 35.7% (35/98) having mild, moderate, and severe thrombocytopenia, respectively (Table 2 and Table 3). The mean PLT count was 76.90 ± 115.05 K/μL, which was beyond the lower limit of the reference interval and considered to be in the range of moderate thrombocytopenia (Table 2). The second most observed hematological alteration was anemia (89/108, 82.4%), with 23.6 % (21/89), 24.7% (22/89), and 51.7% (46/89) having mild, moderate, and severe anemia, respectively (Table 2 and Table 3). None had erythrocytosis. The mean HCT was 26.18 ± 12.65%, which was below the lower limit of the reference interval and fell into the range of moderate anemia (Table 2). The most common combinations were severe anemia with moderate thrombocytopenia (28/108, 25.9%), and severe anemia with severe thrombocytopenia (17/108, 15.7%) (Table 3). None had normal HCT with thrombocytosis, severe anemia with thrombocytosis, or severe anemia with normal PLT (Table 3).

Reticulocytosis was observed in 52.8% (57/108) of the cases, with mean RETIC being 115.28 ± 116.14 K/μL. Microcytic, normocytic, and macrocytic features were seen in 30.6% (33/108), 48.1% (52/108), and 21.3% (23/108) of the cases, respectively. Hypochromic, normochromic, and hyperchromic features were seen in 17.6% (19/108), 75.0% (81/108), and 7.40% (8/108) of the cases, respectively (Table 2).

The most frequent leukogram change was monocytosis (70/108, 64.8%), with a mean monocyte count of 1.81 ± 1.37 K/μL, which is above the reference interval. None had monocytopenia. The second most common leukogram change was eosinopenia (44/108, 40.7%). Other alterations included leukocytosis (24/108, 22.2%), neutrophilia (21/108, 19.4%), lymphopenia (19/108, 17.6%), neutropenia (13/108, 12.0%), leukopenia (12/108, 11.1%), lymphocytosis (9/108, 8.3%), and basophilia (8/108, 7.4%). Normal ranges of WBC, NEU, LYM, and BASO were observed in 66.7% (72/108), 68.5% (74/108), 74.1% (80/108), and 92.6% (100/108) of the cases, respectively (Table 2).

Table 4 shows a summary of descriptive statistics of coagulation time results, with 17/108 (15.7%) and 19/108 (17.6%) having APTT and PT results, respectively. Among the 17 cases with APTT results, 10/17 (58.8%) had prolonged APTT, and mean APTT was 108.3 ± 27.41 seconds, which was mildly prolonged. Among 19 cases with PT results, 16/19 (84.2%) and 3/19 (15.8%) had normal and decreased PT, respectively. None had prolonged PT. Mean PT was 13.1 ± 2.00 seconds, which was within the normal range.

Table 5 shows a summary of descriptive statistics of serum biochemistry results, where the majority had normal BUN, CREA, TP, ALB, ALT, ALP, GGT, and TBIL levels. However, 52% of the dogs had a BUN to CREA ratio > 16, with a mean value of 29.52 mg/dL. Hyperglobulinemia was found in 46.3% (50/108) of the cases. Elevated ALP and GGT were found in 30.6% (33/108) and 21.3% (23/108) of patients, respectively. Mean ALP and GGT were 300.39 ± 713.05 U/L and 12.37 ± 92.20 U/L respectively, which were above the reference range. Other alterations include elevated BUN (19/108, 17,6%), hyperproteinemia (16/108, 14.8%), elevated ALT (12/108, 11.1%), and hyperbilirubinemia (9/108, 8.3%). Additionally, 17/108 (15.7%) had documented cPLI test results. Among them, 10 (58.8%) and 7 (41.2%) had normal and abnormal results, respectively. 

Table 6 shows a summary of statistics on urinalysis parameters where 22/108 (20.4%) had documented urinalysis results. Only 3/22 (13.6%) had hyposthenuria or isosthenuria. Semiquantitative 2+ and 3+ proteinuria were found in 13.6% (3/22) and 31.8% (7/22), respectively, and 2+ and 3+ bilirubinuria were found in 27.2% (6/22) and 31.8% (7/22), respectively.

## 4. Discussion

The present study aims to describe clinical and clinicopathological alterations in dogs infected with *B. gibsoni*, categorizing patterns and magnitude of changes. Identification of the small *B. gibsoni* in blood smears presents a diagnostic challenge for clinicians when low levels of parasitemia are present [26]. Additionally, stain artifacts and other intraerythrocytic inclusions might be hard to differentiate. Although PCR is far more sensitive and specific, its cost and availability outside of referral laboratories is a limitation. This is why it becomes important for clinicians to recognize a cluster of clinical, hematological and biochemical findings that are most consistent with this infection. In this study, we provide evidence to support the use of a cluster of clinical and clinicopathological findings that can raise a clinician’s suspicion index for this infection and prompt early definitive PCR testing.

In this study, thrombocytopenia was observed in the majority of cases, which is consistent with previous findings [12,17]. Notably, most cases exhibited moderate and severe thrombocytopenia. The underlying mechanisms of thrombocytopenia have not been fully elucidated, but several causes have been proposed, including immune-mediated thrombocytopenia, disseminated intravascular coagulation (DIC), infection-related vasculitis leading to platelet consumption, and splenic sequestration and aggregation [22,30,31]. Anemia, resulting from intravascular or more likely extravascular hemolysis [32,33,34], was another significant finding consistent also with other studies [4,12,17]. Mechanisms of hemolysis include mechanical destruction of erythrocytes, release of hemolytic factors by the protozoan, increased osmotic fragility, and immune-mediated destruction [34]. The severity of the observed anemia was predominantly moderate to severe. The most frequent hematological abnormalities were severe anemia coupled with moderate or severe thrombocytopenia. None of the cases had severe anemia with normal platelet count or thrombocytosis, and just 0.9% (1/108) had severe anemia coupled with mild thrombocytopenia. This would suggest that a lack of severe or moderate thrombocytopenia in severely anemic dogs might preclude a *B. gibsoni* infection as the cause of anemia, implying a different etiology. The small number of dogs with normal or increased PLT coupled with milder anemia might represent chronic and subclinical infections. No cases were reported with normal hematocrit and thrombocytosis. The variable magnitudes and combinations of anemia and thrombocytopenia observed in this study hoped to provide novel insights and further guidance for diagnosis of babesiosis, for example, when to prioritize *B. gibsoni* infection in differential diagnoses.

The predominant clinical signs observed in the study were anorexia, pallor, lethargy, pigmenturia, fever, and splenomegaly, which aligns with the findings of previous studies [16,17]. Additionally, vomiting, not often directly associated with babesiosis, was present in almost one quarter of the cases. Despite the higher number of cases with pigmenturia, jaundice, often seen with severe hemolytic anemia, was observed in just over 7% of the dogs in this study. This is likely due to the lower renal threshold for bilirubin in dogs and correlates with a lower proportion of dogs that had high TBIL increases, which is different to what is observed in dogs with infections from other *babesia* species such as *B. rossi* [20]. Jaundice might be present in dogs that had a more acute and significant hemolytic crisis causing prehepatic jaundice or dogs where hepatic hypoxia consequent to severe anemia might have caused hepatocellular or intrahepatic jaundice. In the present study, a likely extravascular hemolysis is supported by the presence of bilirubinuria and the lack of hemoglobinuria, which would be expected with intravascular hemolysis and is common in other species like *B. rossi* and *B. canis* [20,35]. Fever was present in a third of the cases, which is a hallmark of the host cytokine response more common with other *Babesia* species too [20,35,36].

Hematological findings such as regenerative features including spherocytosis, polychromasia, reticulocytosis, anisocytosis, and intraerythrocytic piroplasms were all noted. However, due to the limited number of cases with available blood smear data in our study, further investigation including magnitude of changes is required to substantiate these findings. Although spherocytosis was present in nearly half of the cases that had a blood smear, only a small proportion of cases had positive saline agglutination. These findings might be an indication of an unclear involvement of immune-mediated hemolytic anemia in babesiosis; however, the low number of cases tested, especially for agglutination, limits its interpretation.

While regenerative anemia was observed in half of the cases based on CBC results, it was not a consistent finding across all instances. Macrocytosis and hypochromasia were present in about 20% of dogs, while increased reticulocyte count was present in half of the cases. These variations may be attributed to the presence of pre-regenerative anemia, where bone marrow requires two to three days to respond, and the inclusion of cases at different stages of infection in the study could further contribute to these discrepancies.

Leukogram changes were non-specific. The majority of cases had normal white blood cell counts. The most predominant leukogram alteration was monocytosis, which can either be due to infection or stress response, or be an artefact associated to the use of in-house analyzers, where monocytes can be misidentified instead of band neutrophils [37]. Similar studies have also reported monocytosis in *B. gibsoni* infection [17]. However, we suggest a differential cytology should be performed to differentiate true monocytosis from band neutrophilia.

Coagulopathies triggered by infections from other *Babesia* species have been reported [38,39]. This is one of the few studies analyzing parameters related to secondary hemostasis in dogs infected with *B. gibsoni*. The majority of cases with results had prolonged APTT with normal PT, which may indicate that inflammation and a consequent DIC in a subsection of cases might be involved in the pathogenesis [40]. Similar findings were observed in a report of dogs infected with larger forms of *Babesia*, where there were more dogs with prolonged APTT than those with prolonged PT [41]. Further investigation involving larger sampling and the incorporation of sensitive parameters for DIC, such as fibrin/fibrinogen degradation products and D-dimers, could provide more insights into the impact of *B. gibsoni* infection on secondary hemostasis.

Most serum biochemistry parameters fell within normal ranges. Azotemia was not as commonly seen as other abnormalities, which is consistent with findings reported previously [17,42]. The BUN:CREA ratio was, however, above the normal range, and that is a feature that has been observed in other *Babesia* infections supporting a non-renal origin [43]. Hyperglobulinemia was present in nearly half of the cases, which is in line with several other studies [5,17,30,44]. This condition is presumably due to production of an acute phase protein as a result of a systemic inflammatory response [45]. Albumin levels were generally elevated or within normal limits, corroborating the results of other studies [22,30,46,47]. Hypoalbuminemia may be caused by protein-losing nephropathy due to glomerulonephritis and acute renal injury subsequent to a *B. gibsoni* infection [45], or it can occur when there is a decrease in albumin production, a negative acute-phase protein, during systemic inflammatory response [45]. The majority of cases had normal liver enzyme ranges, aligning with previous studies [42,44]. Elevated ALP and GGT levels were the most predominant alterations observed in liver parameters, aligning with those of other studies [16,48]. The proposed mechanisms for elevated ALP are cholestasis and an increase in endogenous glucocorticoids due to illness [30]. In conclusion, serum biochemistry results often lie within normal intervals, providing little help in the diagnosis of a *B. gibsoni* infection.

Proteinuria and bilirubinuria were common findings of urinalysis, but have only been reported anecdotally on various case reports [5,22,25,46]. Pigmenturia was a common clinical sign in 35.2% of dogs. Although this would be observed with hematuria, hemoglobinuria and bilirubinuria, bilirubinuria was the only common finding. This may be attributed to cholestatic hepatobiliary disease and extravascular hemolysis, most likely in this case [49]. Urine sediment was not commonly reported, therefore hematuria or the lack of it could not be confirmed. Proteinuria may be due to immune-mediated glomerular injury [50]; however, a high proportion of patients in the present study were non-azotemic. Further investigation of the effect of *B. gibsoni* infection on kidneys is suggested. This would be better demonstrated with the urine protein-to-creatinine ratio (UPC).

The aim of this study was to enhance our understanding of clinical and clinicopathological alternations associated with *B. gibsoni* infection, thereby adding further diagnostic components in its diagnosis. By describing these clinicopathological alterations, further evidence might be gained to support *B. gibsoni* infection inclusion as a differential diagnosis. However, several limitations exist in this study. Being a retrospective study, it depended heavily on the completeness and quality of medical records. The absence of data, along with variability in descriptions and subjective assessment by different clinicians, may have impacted the consistency and accuracy of findings. Additionally, the absence of a control group may limit the ability to establish a direct causative relationship between infection and observed alterations. Moreover, the study population was sourced from a single emergency and specialty hospital, which may limit the generalizability of results to other geographical locations or different types of veterinary practices.

The data were collected only from cases that met the inclusion criteria. Cases with incomplete biochemistry, hematology, out of the age range, or with comorbidities were excluded from the study. This might have created a bias component in the selection of cases, meaning that we cannot draw firm conclusions about how common the various observations are across “all” *B. gibsoni*-infected dogs. Since the collected data were primarily at presentation, it did not include a follow-up of the cases, classification of complicated versus non-complicated infections, or fatality rate. The severity of the disease and clinical signs were, however, correlated with the severity of anemia. Further studies to address these correlations in depth are warranted. The study design excluded older and younger dogs and reduced the number of samples assessed. The design aimed to decrease interference with the interpretation of values by potential degenerative and chronic comorbidities of older dogs. This decision was additionally supported by the fact that babesiosis is less prevalent in older dogs in Hong Kong [12]. Some examples of the comorbidities in older dogs that would affect interpretation include cases of hepatitis, gallbladder mucocele, cardiomyopathy, chronic kidney disease, mast cell tumor, lymphoma, testicular neoplasia, hyperadrenocorticism, among others. Although young dogs might suffer a more severe form of the disease that would then not be reported in this study; this occurrence is rare [14]. Additionally, younger puppies might have immature immune, cardiovascular, and hematopoietic systems as well as decreased renal and hepatobiliary function [51,52]. Future studies should be designed as prospective case-control studies, sourcing cases from more than a single hospital to make the findings more representative of the target population and reduce biases that may arise from the specific practices, patient demographics, or disease prevalence peculiar to a single hospital.

In conclusion, the present study describes the occurrence of clinical signs as well as the alterations and magnitude of changes in hematological, serum biochemical, and urinalysis data in a cohort of selected dogs infected with *B. gibsoni* in Hong Kong. This is one of the few studies that analyzed blood smear evaluation, coagulation times, and urinalysis which may help in guiding the decision for diagnostics. This study adds to the body of knowledge, and its findings can be extrapolated to other regions where *B. gisoni* is also present.

## 5. Conclusions

This study provides valuable insights into the clinical signs and clinicopathological alterations associated with *B. gibsoni* infection in dogs in Hong Kong, the most common arthropod-borne infection in dogs. The most frequent clinical signs were anorexia, pallor, lethargy, pigmenturia, and fever, whereas the most frequent hematological alterations were thrombocytopenia and anemia, with the most common combination being severe anemia with moderate or severe thrombocytopenia. White blood cell counts varied and were non-specific. Prolonged APTT without prolonged PT was also observed, although the sample size was small. Most serum biochemistry parameters were within normal ranges, but hyperglobulinemia and BUN:CREA ratio increases were detected in approximately half of the cases. Proteinuria and bilirubinuria were common findings in the urinalysis. These findings help in recognizing consistent patterns in clinical signs, hematology, serum biochemistry, and urinalysis, supporting an early presumptive diagnosis and prompting confirmatory testing, such as polymerase chain reaction (PCR). Early detection of babesiosis can lead to timely intervention, improving the treatment outcome in affected dogs.

## Figures and Tables

**Table 1 animals-15-00645-t001:** Frequency and percentage of presenting signs and physical examination findings among the sampled dogs (n = 108).

Presenting Signs and Physical Examination Findings	Frequency (N = 108)	Percentage (%)
Anorexia	73	67.6
Pallor	57	52.7
Lethargy	49	45.4
Pigmenturia	38	35.2
Fever	36	33.3
Splenomegaly	27	25.0
Vomitting	26	24.1
Diarrhea	16	14.8
Weakness	15	13.9
Jaundice	8	7.4

**Table 2 animals-15-00645-t002:** CBC parameters: mean values, standard deviations (SD), minimum (Min), maximum (Max), and number of observed cases (n) and percentage of cases (%) categorized as decreased, normal, or increased with respect to the reference range.

Parameter	Reference Range	Unit	Mean ± SD	Min	Max	Decreased n (%)	Normal n (%)	Increased n (%)
**HCT**	37.3–61.7	%	26.18 ± 12.65	5.9	60.3	89 (82.4%)	19 (17.6%)	0 (0.00%)
**MCV**	61.6–73.5	fL	66.71 ± 9.76	44.9	94.1	33 (30.6%)	52 (48.1%)	23 (21.3%)
**MCHC**	32.0–37.9	g/dL	33.83 ± 3.48	18.4	41.4	19 (17.6%)	81 (75.0%)	8 (7.40%)
**RETIC**	10.0–60.0	K/μL	115.28 ± 116.14	2.5	461.5	9 (8.3%)	42 (38.9%)	57 (52.8%)
**WBC**	5.05–16.76	K/μL	13.39 ± 8.78	1.66	59.16	12 (11.1%)	72 (66.7%)	24 (22.2%)
**NEU**	2.95–11.64	K/μL	8.51 ± 6.98	0.07	41.12	13 (12.0%)	74 (68.5%)	21 (19.4%)
**LYM**	1.05–5.10	K/μL	2.88 ± 2.08	0.26	13.47	19 (17.6%)	80 (74.1%)	9 (8.3%)
**MONO**	0.16–1.12	K/μL	1.81 ± 1.37	0.21	6.48	0 (0.00%)	38 (35.2%)	70 (64.8%)
**EOS**	0.06–1.23	K/μL	0.16 ± 0.28	0	2.17	44 (40.7%)	63 (58.3%)	1 (0.90%)
**BASO**	0.00–0.10	K/μL	0.03 ± 0.07	0	0.47	0 (0.00%)	100 (92.6%)	8 (7.4%)
**PLT**	148–484	K/μL	76.90 ± 115.05	0	906	98 (90.7%)	8 (7.4%)	2 (1.9%)

HCT—hematocrit, MCV—mean cell volume, MCHC—mean cell hemoglobin concentration, RETIC—reticulocyte count, WBC—white blood cells, NEU—neutrophils, LYM—lymphocytes, MONO—monocytes, EOS—eosinophils, BASO—basophils, PLT—platelets. Normal distribution for all parameters other than RETIC and PLT.

**Table 3 animals-15-00645-t003:** Cross-tabulation of case distribution based on severity of anemia and thrombocytopenia in 108 dogs.

	Severe Anemia	Moderate Anemia	Mild Anemia	Normal HCT
**Severe thrombocytopenia**	17 (15.7%)	8 (7.4%)	4 (3.7%)	6 (5.6%)
**Moderate thrombocytopenia**	28 (25.9%)	8 (7.4%)	10 (9.3%)	6 (5.6%)
**Mild thrombocytopenia**	1 (0.9%)	3 (2.8%)	1 (0.9%)	6 (5.6%)
**Normal PLT**	0 (0.00%)	2 (25.0%)	5 (4.6%)	1 (0.9%)
**Thrombocytosis**	0 (0.00%)	1 (0.9%)	1 (0.9%)	0 (0.00%)

HCT—hematocrit, PLT—platelets. Number of animals out of 108 and respective percentage for each degree of severity n (%).

**Table 4 animals-15-00645-t004:** Coagulation times: total numbers (N), mean values, standard deviations (SD), minimum (Min), maximum (Max), and number of observed cases (n) and percentage (%) categorized as decreased, normal, or increased with respect to the reference range.

Parameter	Reference Range	Unit	Total (N)	Mean ± SD	Min	Max	Decreased n (%)	Normal n (%)	Increased n (%)
**APTT**	72–102	second	17	108.3 ± 27.41	55	169	1 (5.9%)	6 (35.3%)	10 (58.8%)
**PT**	11–17	second	19	13.1 ± 2.00	9	16	3 (15.8%)	16 (84.2%)	0 (0.00%)

APTT—activated partial thromboplastin time on citrated blood, PT—prothrombin time on citrated blood. Normal distribution for all parameters.

**Table 5 animals-15-00645-t005:** Serum biochemistry parameters: mean values, standard deviations (SD), minimum (Min), maximum (Max), and number of observed cases (n) and percentage of cases (%) categorized as decreased, normal, or increased with respect to reference range.

Parameter	Reference Range	Unit	Mean ± SD	Min	Max	Decreased n (%)	Normal n (%)	Increased n (%)
**BUN**	2.5–9.6	mmol/L	7.59 ± 7.16	1.1	46.4	7 (6.5%)	82 (75.9%)	19 (17.6%)
**CREA**	44–159	µmol/L	85.12 ± 67.13	23	534	10 (9.3%)	92 (85.2%)	6 (5.6%)
**TP**	52–82	g/L	71.58 ± 12.71	26	115	6 (5.6%)	86 (79.6%)	16 (14.8%)
**ALB**	23–40	g/L	27.34 ± 4.98	15	44	15 (13.9%)	91 (84.3%)	2 (1.9%)
**GLB**	25–45	g/L	44.81 ± 10.51	12	85	1 (0.9%)	57 (52.8%)	50 (46.3%)
**ALT**	10–125	U/L	89.42 ± 151.23	10	1043	0 (0.0%)	96 (88.9%)	12 (11.1%)
**ALP**	23–212	U/L	300.39 ± 713.05	15	5767	4 (3.7%)	71 (65.7%)	33 (30.6%)
**GGT**	0–7	U/L	12.37 ± 92.20	0	952	0 (0.0%)	85 (78.7%)	23 (21.3%)
**TBIL**	0–15	µmol/L	27.34 ± 92.24	0	809	0 (0.0%)	99 (91.7%)	9 (8.3%)

BUN—blood urea nitrogen, CREA—creatinine, TP—total protein, ALB—albumin, GLB—globulin, ALT—alanine aminotransferase, ALP—alkaline phosphatase, GGT—γ-glutamyl transpeptidase, TBIL—total bilirubin. Normal distribution other than ALT, ALP, GGT, and TBIL

**Table 6 animals-15-00645-t006:** Descriptive statistics of urinalysis results, including the frequency and percentage of each finding.

Finding	Frequency (N = 22)	Percentage (%)
**Hyposthenuria/isosthenuria**	3	13.6
**Urine protein**		
Negative	5	22.7
Trace	1	4.5
1+	6	27.3
2+	3	13.6
3+	7	31.8
**Urine bilirubin**		
Negative	7	31.8
1+	2	9.1
2+	6	27.3
3+	7	31.8

## Data Availability

The data from this study are available upon request from the corresponding author, in accordance with patient confidentiality and hospital policies.

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
