# Peer review of "Clinical and Clinicopathological Features of 108 Dogs Infected with Babesia gibsoni in Hong Kong"

_animals, 2025, doi:10.3390/ani15050645_

Round 1

Reviewer 1 Report

Comments and Suggestions for Authors

The study has several limitations reported in the discussion. I understand that the shortcomings of the study are largely incorrigible, but I am inclined to recommend this manuscript for publication, due to the still too little information about B. gibsoni infection and I believe that every data seems necessary and valuable. In the manuscript there are some flaws to be assessed before it can be published.

Specific comments:

Title

The authors should report in the title that dogs are from Hong Kong

Introduction

Page 1, Line 35: The authors should at least report the Order and Phylum to wich Babesia gibsoni belong

Page 1, Line 38: Change H. longicornis in Haemaphysalis longicornis

Page 2, line 44: the authors should specific the genotype of Babesia gibsoni

Page 2, Line 45:  indicate the dog breeds in which the presence of B. gibsoni is most frequently reported and which are the most important risk factors

 Material and methods

Page 3, Line 102-112: The authors should eliminate this part and specify the meaning of the acronyms in the table captions or in the results

 Discussion

Page 8, Line 195: specify what means DIC

Author Response

Comments 1: The authors should report in the title that dogs are from Hong Kong

Response 1: Thank you for pointing this out. We agree with this comment. Therefore, Hong Kong is added in the title.

Comments 2: Page 1, Line 35: The authors should at least report the Order and Phylum to which Babesia gibsoni belong

Response 2: Thank you for pointing this out. “B. gibsoni, order Piroplasmida, Class Sporozoasida, and phylum Apicomplexa” added

Comments 3: Page 1, Line 38: Change H. longicornis in Haemaphysalis longicornis

Response 3: Thank you for pointing this out. We agree with this comment. Therefore, H. longicornis is changed to Haemaphysalis longicornis

Comments 4: Page 2, line 44: the authors should specific the genotype of Babesia gibsoni

Response 4: Genotype is B. gibsoni (Asian genotype), added.

Comments 5: Page 2, Line 45: indicate the dog breeds in which the presence of B. gibsoni is most frequently reported and which are the most important risk factors

Response 5: Thank you for pointing this out. Risk factors for infection including mixed breed dogs, of age below 10 years old and pure breed for horizontal transmissions via dog fights were added

Comments 6: Page 3, Line 102-112: The authors should eliminate this part and specify the meaning of the acronyms in the table captions or in the results

Response 6: Thank you for pointing this out. We agree with this comment. Therefore, this section is deleted, and the acronyms are specified in the table captions in tables 2, 5, 6, 7.                                                                                         

Comments 7: Page 8, Line 195: specify what means DIC

Response 7: Meaning has been added

Reviewer 2 Report

Comments and Suggestions for Authors

Review: animals-3423799

Clinicopathological features of 108 dogs infected with Babesia gibsoni

The authors analyzed the clinical records of 108 dogs that met their inclusion criteria of infection with Babesia gibsoni and provided a description of the disease clinically and clinicopathologically. Reports like this are valuable as the best way to get a birds eye view of a disease is to study large numbers of cases in a cohort – and this is what these investigators have tried to do. I do believe the paper has merits but there are some serious concerns that I think need to be addressed.

My general comments follow:

The study was retrospective and included cases over a 10 year span. There is no indication of whether the cases were consecutive or not, and hence it seems very likely that there is a bias in the kind of cases that were included in the study, which will very likely be reflected in a bias in the data. For example, most cases were anemic but if all cases that presented with the diagnosis of B. gibsoni, and not just a random selection of cases that met the current inclusion criteria, would anemia and the spread of the degree of anemia have remained the same? Were the cases selected sick enough to require hospitalization? Were they all seen on an outpatient basis? Was the data collected before or after treatment (and what treatment, including supportive treatments like blood transfusions), if anemia prompted a PCR test for B. gibsoni then it is little surprise that most were anemic. To get a true picture of the disease and the relative frequency of particular presentations, consecutive cases need to be collected to ensure a true representation of the various findings.

There is no attempt to help the reader understand from the data what would make a case severe (complicated). What proportion of the cases described were severe (complicated) – and what made them severe (complicated). What proportion were mild (uncomplicated). There is no indication of outcome. What proportion of these cases survived? Required advanced hospital care?

It is said that the most severe form of the disease is seen in young puppies (line 48) and yet this age group was excluded. Age plays an important role in the form of disease that presents and it seems odd that only cases 1 – 9 years of age were included. In other canine babesia infections, age plays a role in the form of disease seen.

What was done to try to ensure that dogs with B. gibsoni only had this single tick-borne pathogen? Co-infections with other parasites are not uncommon and do influence the clinical picture seen.

There are several places where the English language needs an edit.

Some specific comments include:

Line 42: Is there a ref for B. gibsoni being found locally transmitted in Africa? I am not – although imported cases have been reported it is not an indigenous disease in Africa as far as I am aware.

Line 48: Ref 12 does not refer to B. gibsoni. Although the reference calls the parasite B. canis, it is clear now that in fact it was B. rossi – which results in a much more severe disease more commonly than what B. gibsoni does.

Line 57-58: Ref 2 is a poor ref to use here. You need to go back to an original (not a review) report that describes obvious organ dysfunction in B. gibsoni (I’d be interested in knowing if you could find one!).

Line 60 – 61: Ref 17 again describes the disease caused by B. rossi (quite a different disease caused by B. gibsoni). Ref 12 is also a review – it is always best to refer to original work and not reviews for statements like this. 

It would be great if you could give an indication of the sensitivity of a peripheral blood smear vs. a PCR for the diagnosis. Why is there relatively little reliance on a blood smear for this infection when for other species of this same genus, a blood smear is the go-to diagnostic test.

Is a slide review not standard practice when a complete blood count is performed? Surely in an area where there is a significant case load of a tick borne pathogen, a slide review looking for the parasite should be standard practice? This would also be true to evaluate red cell morphology. How was the spherocyte percentage determined?

Paragraph beginning with line 102: the various cells counted in the CBC are listed but reticulocytes are not noted. How was the reticulocyte count established?

Line 107: ‘citrate prothrombin time’ should read ‘prothrombin time on citrated blood (same for the PTT).

The method(s) used for the urinalysis are not made clear in this section of the M&M. Was there no evidence of hemoglobinuria? It would seem to me that this is a disease characterized by extravascular hemolysis more than intravascular hemolysis? Comment on this please as other canine Babesia parasites cause very obvious intravascular hemolysis and hemoglobinuria.

Is ref 22 a dog-specific reference?

In the section on Statistical Analysis, no mention is made of whether the data was tested for normality – and yet means and SD’s are reported.

Line124: do you have any data on if the dogs were sterilized or not?

Line 125: reporting the number of dogs of a certain breed are affected is not helpful if we have no idea about how common that breed is in the general population. If most dogs in the general population are mongrels then it is probably not surprising that most affected dogs are mongrels. I do not believe figure 1 is useful (it should be removed).

Table 1: It is very interesting that fever was not a more common presenting clinical sign. Fever is a hallmark of a host cytokine response. This is something that should be discussed in the Discussion as it probably relates to why the disease caused by B. gibsoni is less severe than that caused by B rossi and B canis. I found it interesting that there was so little comparison made between the disease caused by B. gibsoni and that caused by other canine Babesia parasites.

Table 2 and related text: it is not necessary to report on HCT, RBCC and HGB – they all tell us basically the same thing. Typically a clinician is most interested in the HCT.

Table 4 should be removed. The text discussion is adequate.

Paragraph starting with line 161: there is no indication of if there was an increase in band cells (left shift).

In other hemolytic diseases (such as IMHA and the disease caused by B. rossi), there is a disproportionate increase of urea to creatinine. Was this true here?

Line 222: what I an intraerythrocytic inclusion? A parasite? A Heinz body? Nuclear remnant?

Is there any sense of how long these dogs had been ill before being presented for care? The point is made that there may not have been enough time for the marrow to cause signs of red cell regeneration. It has been shown that in B. rossi the anemia (despite being obviously hemolytic) is inappropriately regenerative.

References: There are numerous references where Babesia Gibsoni should read Babesia gibsoni or where scientific names are not in italics.

Comments on the Quality of English Language

There are areas in the manuscript where the English language needs an edit. 

Author Response

Comments 1: The study was retrospective and included cases over a 10 year span. There is no indication of whether the cases were consecutive or not, and hence it seems very likely that there is a bias in the kind of cases that were included in the study, which will very likely be reflected in a bias in the data. For example, most cases were anemic but if all cases that presented with the diagnosis of B. gibsoni, and not just a random selection of cases that met the current inclusion criteria, would anemia and the spread of the degree of anemia have remained the same? Were the cases selected sick enough to require hospitalization? Were they all seen on an outpatient basis? Was the data collected before or after treatment (and what treatment, including supportive treatments like blood transfusions), if anemia prompted a PCR test for B. gibsoni then it is little surprise that most were anemic. To get a true picture of the disease and the relative frequency of particular presentations, consecutive cases need to be collected to ensure a true representation of the various findings.

Response 1: Expanding over 10 years we initially obtained raw data for 264 dogs with a PCR diagnosis. Then due to missing data, age range selection…etc we excluded 156 dogs. We agree we only included dogs that had been diagnosed by PCR and for that reason this is not a representation of “all” dogs infected with B. gibsoni but includes “all” dogs that came to the clinic for any reason and were diagnosed by PCR and had the data we could analyze.

All the data was collected on presentation, hence, before treatment. Most presented clinical signs like lethargy and anorexia, some had had ticks and that’s why they came and were tested. Surely the ones that presented with anemia and thrombocytopenia, fever, splenomegaly...etc would be suspicious of infection and tested. Dogs coming for routine visits or procedures that had the needed data were also included though and they would represent chronic or subclinical undiagnosed infections.

Specific and supportive treatment depended on the case and included antiparasitic protocols and fluid therapy and blood products in cases where it was deemed necessary. In this study we aim to provide diagnostic findings information (rather than epidemiologic, therapeutic…) and add to the body of knowledge. We acknowledge there were many more dogs infected that were excluded or not diagnosed. The selection criteria aimed to standardize comparable data sets that would be more genuine, with less comorbidities or limited immune response due to age, as it would be much harder to interpret results otherwise. We agree the design of the study (being retrospective) is a limitation. We appreciate your comments and believe they will encourage us to conduct further focused studies. We have expanded the discussion to include these limitations and acknowledge the need of studying consecutive cases and of prospective designs.

Comments 2: There is no attempt to help the reader understand from the data what would make a case severe (complicated). What proportion of the cases described were severe (complicated) – and what made them severe (complicated). What proportion were mild (uncomplicated). There is no indication of outcome. What proportion of these cases survived? Required advanced hospital care?

Response 2: Thank you for your comment. We focused the aim of the study mostly in describing the most common clinical findings and the diagnostic findings. We did not collect subsequent data. There were 17 dogs with more severe anemia and TCP and 28 with marked anemia and moderate TCP. These two groups were the dogs that showed worse clinical signs. We have not investigated however whether there were fatalities. Out of the 108, none died on the day of presentation. Our aim is to guide clinicians towards the diagnosis rather than the classification of complicated. We surely had some more complicated unusual cases I can even remember from treating myself but many of them as it occurs in B. gibsoni were milder and would respond to therapy. I added some info in the discussion.

Comments 3: It is said that the most severe form of the disease is seen in young puppies (line 48) and yet this age group was excluded. Age plays an important role in the form of disease that presents and it seems odd that only cases 1 – 9 years of age were included. In other canine babesia infections, age plays a role in the form of disease seen.

Response 3: We know that age is a risk factor and in a large study we found infections to be less common in dogs <10 years old (Muguiro et al. 2023). Additionally, we also know that although it’s true young puppies suffer a hyperacute or more severe presentation, it is also true this is a rare occurrence (Boozer et al 2005). We aimed to present the data of the most commonly affected age range, that would have the least interference with either chronic disease/comorbidities in older patients, or with the lack of maturity of; immune system, renal and hepatic function, blood pressure...etc in younger puppies making them in first place more likely to suffer complicated forms of this or any other disease. This would make a very interesting study focusing on young puppies alone.

Looking at the raw data the amount of comorbidities in older patients was remarkable and that’d have made the interpretation rather difficult.

Comments 4: What was done to try to ensure that dogs with B. gibsoni only had this single tick-borne pathogen? Co-infections with other parasites are not uncommon and do influence the clinical picture seen.

Response 4: Our PCR tests are all performed at CityU VDL (since July 2018) and previously at Pathlab diagnostics. Their tick fever panels include Babesia spp., B. gibsoni, B. canis and Ehrlichia canis. Before submitting the PCR, most will have an in-house test (Snap Iddexx 4DX including HW, EC, AP and Lyme). However, it is true not all cases might have had the full panels as being a retrospective study the full reports are not always attached. It is common practice; we could not even run B. gibsoni alone until very recently. We have included this information in the limitations.

There were 2 anaplasma +ve and 4 EC positive on serology. One had all 4DX positive results which were false positives for HW and for EC afterwards. All of them had mild or normal HCT and the disease was mild in presentation too. The diagnosis of babesiosis was always by a PCR panel as included in the manuscript.

Comments 5: There are several places where the English language needs an edit.

Response 5: We have edited extensively the manuscript to improve the language.

Comments 6: Line 42: Is there a ref for B. gibsoni being found locally transmitted in Africa? I am not – although imported cases have been reported it is not an indigenous disease in Africa as far as I am aware.

Response 6: Prof Schoeman has reported it as endemic in North and East Africa (Schoeman 2009).

Comments 7: Line 48: Ref 12 does not refer to B. gibsoni. Although the reference calls the parasite B. canis, it is clear now that in fact it was B. rossi – which results in a much more severe disease more commonly than what B. gibsoni does.

Response 7: I updated the reference (Birkenheuer et al 2005) that reported fighting dogs suffered from this, the sentence has been rephrased. We have occasionally seen this type of presentations here too with DIC and SIRS published as a case report and there are other isolated reports too that I have added. 

Comments 8: Line 57-58: Ref 2 is a poor ref to use here. You need to go back to an original (not a review) report that describes obvious organ dysfunction in B. gibsoni (I’d be interested in knowing if you could find one!).

Response 8: I agree with you that B. gibsoni is less commonly associated with organ dysfunction or at least in not widely reported as it happens with B. rossi. The sentence has been rephrased, and the references have been changed. There are several reports, early back to Conrad in 1991 to more recently in 2021 with complications including DIC, glomerulonephropathy, renal failure…etc. We have rephrased to reflect that rather than organ failure.

Comments 9: Line 60 – 61: Ref 17 again describes the disease caused by B. rossi (quite a different disease caused by B. gibsoni). Ref 12 is also a review – it is always best to refer to original work and not reviews for statements like this.

Response 9: The sentence has been rephrased and the references updated as per comment 8.  

Comments 10: It would be great if you could give an indication of the sensitivity of a peripheral blood smear vs. a PCR for the diagnosis. Why is there relatively little reliance on a blood smear for this infection when for other species of this same genus, a blood smear is the go-to diagnostic test.

Response 10: PCR has been reported to be very sensitivite, detecting DNA in 2.5 ul of blood with parasitemia of 0.000002% (Fukumoto 2001). However, the presence of piroplasm inside red blood cells by and experienced veterinarian or pathologist together with supporting signs would be a reliable diagnosis. We would always confirm with PCR though. B. gibsoni is very small and on many occasions, there are artifacts that can make its diagnosis difficult, especially when done in-house. Additionally, there are also other hemoparasites that could be mistakenly misdiagnosed; hence confirmation is by PCR, far more sensitive and specific. Additionally, in a retrospective study like ours with the analytics performed mostly in house there might be a large variability of operator skills when interpreting cytology if this was the only diagnostic method. 

Comments 11: Is a slide review not standard practice when a complete blood count is performed? Surely in an area where there is a significant case load of a tick-borne pathogen, a slide review looking for the parasite should be standard practice? This would also be true to evaluate red cell morphology. How was the spherocyte percentage determined?

Response 11: We agree it would be ideal to run a blood smear after every CBC but unfortunately that is not the case in practice due to time restrains in many cases. This is another limitation that we mentioned due to the study design. Hence, the lower number of dogs that had a blood smear in the study. The smears were reviewed either by the clinicians in-house (like when the dog presented anemic to the ER) or by pathologists when sent to a referral lab for hematology.    

Comments 12: Paragraph beginning with line 102: the various cells counted in the CBC are listed but reticulocytes are not noted. How was the reticulocyte count established?

Response 12: Sorry. We had the reticulocyte count too counted by the hematology analyzer and is in the same paragraph following MCHC. However, we have deleted that paragraph following another reviewer comments as appeared redundant and in the paragraph above we mention we had CBC results and all of them have been now added to the table captions and appeared already in the results section.

Comments 13: Line 107: ‘citrate prothrombin time’ should read ‘prothrombin time on citrated blood (same for the PTT).

Response 13: Thank you for pointing this out. “APTT activated partial thromboplastin time on citrated blood, PT prothrombin time on citrated blood” was added in the caption of Table 5.

Comments 14: The method(s) used for the urinalysis are not made clear in this section of the M&M. Was there no evidence of hemoglobinuria? It would seem to me that this is a disease characterized by extravascular hemolysis more than intravascular hemolysis? Comment on this please as other canine Babesia parasites cause very obvious intravascular hemolysis and hemoglobinuria.

Response 14: Hemoglobinuria was not common and was described in the clinical notes meaning red/dark urine. This was actually pigmenturia that would include hematuria, hemoglobinuria and biliribinuria, which was actually the most common finding on urine tests. This would support extravascular hemolysis being more common. The sediment exam of urine was not reported in most cases hence we cannot report whether there were red cells in urine.  

Comments 15: Is ref 22 a dog-specific reference?

Response 15: Yes, it was. We have deleted and changed for a canine reference. Cornell eClinpath actually classifies TCP either subjectively or using human guidelines. Hence, we defined the ranges based on what are normal ranges in veterinary literature.

Comments 16: In the section on Statistical Analysis, no mention is made of whether the data was tested for normality – and yet means and SD’s are reported.

Response 16: We did test for normality (by skewness) but some of them have normal distribution while some do not. Means and SDs should only be reported in those with normal distribution.

Comments 17: Line124: do you have any data on if the dogs were sterilized or not?

Response 17: Thank you for pointing this out. We have the relevant information, which is added

Comments 18: Line 125: reporting the number of dogs of a certain breed are affected is not helpful if we have no idea about how common that breed is in the general population. If most dogs in the general population are mongrels then it is probably not surprising that most affected dogs are mongrels. I do not believe figure 1 is useful (it should be removed).

Response 18: Thank you for pointing this out. Figure 1 and text reporting breeds are removed.

Comments 19: Table 1: It is very interesting that fever was not a more common presenting clinical sign. Fever is a hallmark of a host cytokine response. This is something that should be discussed in the Discussion as it probably relates to why the disease caused by B. gibsoni is less severe than that caused by B rossi and B canis. I found it interesting that there was so little comparison made between the disease caused by B. gibsoni and that caused by other canine Babesia parasites.

Response 19: We have included this in the introduction and the discussion comparing with other babesias and adding references.

Comments 20: Table 2 and related text: it is not necessary to report on HCT, RBCC and HGB – they all tell us basically the same thing. Typically a clinician is most interested in the HCT.

Response 20: Thank you for pointing this out. Rows of data and text reporting RBC and HGB are deleted.

Comments 21: Table 4 should be removed. The text discussion is adequate.

Response 21: We appreciate we have a number of tables but we still believe it is a very easy and graphical way to see the correlation of HCT and PLT in this disease that is rather unique. In the sense that we hardly encounter a severely anemic dog with normal PLT count in B. gibsoni infections. To the point we would seriously consider other differentials rather than B. gibsoni as the cause of anemia.

Comments 22: Paragraph starting with line 161: there is no indication of if there was an increase in band cells (left shift).

We did not had that parameter in the in-house analyzer but we acknowledge that monocytosis can be a common iatrogenic finding when using automatic analyzers when these could be in fact band neutrophils. We see that often in dogs receiving chemotherapy that appear neutropenic when in fact the bone marrow rebounded and are neutrophilic with false monocytosis recorded by the automatic measurement. This would need to be differentiated in further research but monocytosis has been reported in other similar studies assessing B. gibsoni clinicopathological features (possibly for the same reason) and is indeed what general practitioners will see in their analyzers even if iatrogenic. We suggest monocytosis might be falsely elevated when actually are band neutrophils. We included this in the manuscript.

Comments 23: In other hemolytic diseases (such as IMHA and the disease caused by B. rossi), there is a disproportionate increase of urea to creatinine. Was this true here? 

We had not calculated but we did after your comment and the results showed that 52% had BUN:CREA elevated. We have reflected that in the manuscript.

Comments 24: what I an intraerythrocytic inclusion? A parasite? A Heinz body? Nuclear remnant?

Response 24: Thank you for pointing this out. “Intraerythrocytic inclusion” is modified to “intraerythrocytic piroplasms”

Comments 25: Is there any sense of how long these dogs had been ill before being presented for care? The point is made that there may not have been enough time for the marrow to cause signs of red cell regeneration. It has been shown that in B. rossi the anemia (despite being obviously hemolytic) is inappropriately regenerative.

Response 25: The majority had normocytosis, normochromasia but reticulocytosis. In general the blood test was taken at presentation hence when they first came to the clinic. Since that would be early in the onset of the disease in most cases it is feasible more regenerative features were not present yet. 

Comments 26: References: There are numerous references where Babesia gibsoni should read Babesia gibsoni or where scientific names are not in italics.

Response 26: Thank you for pointing this out. Scientific names are modified according to binomial nomenclature.

Reviewer 3 Report

Comments and Suggestions for Authors

In this study, the authors described a retrospective review of the clinicopathological features of dogs infected with Babesia gibsoni  in Hong Kong.

Althought the English text is very well written i have some main concerns:

Althought Babesia gibsoni is the most common arthropod-borne  disease within canine population in Hong Kong, I would suggest authors to include other Babesia species infection. Indeed Babesia vogeli was  described in dogs in same area

In the Result section there are repetitions of data that are described both in tables and in the text. The MS as it is -  looks redundant. 

Statistical analysis is described in M&M sections - but data in the result section  are not clearly statistically supported. 

I would suggest authors to organize the data and provide a table summarizing the key clinical outcomes to enhance the readability of the document. 

Author Response

We have edited extensively the manuscript to improve language.

We have added comparison with other species of babesia species and modified the material and methods and results sections. We have also changed some of the tables with redundant information and added to the discussion.

Thank you for your comments.

Round 2

Reviewer 2 Report

Comments and Suggestions for Authors

Review #2: Title: Clinicopathological features of 108 dogs infected with Babesia gibsoni
Journal: Animals

B gibsoni case series review

I think this is a helpful piece of work and adds valuable data to the canine Babesia literature. The authors are to be encouraged for the investigation, and I would encourage them to carefully consider my further comments below.

I think the title could change to “Clinical and clinicopathological features of 108 dogs….”

Comments 1: The study was retrospective and included cases over a 10 year span. There is no indication of whether the cases were consecutive or not, and hence it seems very likely that there is a bias in the kind of cases that were included in the study, which will very likely be reflected in a bias in the data. For example, most cases were anemic but if all cases that presented with the diagnosis of B. gibsoni, and not just a random selection of cases that met the current inclusion criteria, would anemia and the spread of the degree of anemia have remained the same? Were the cases selected sick enough to require hospitalization? Were they all seen on an outpatient basis? Was the data collected before or after treatment (and what treatment, including supportive treatments like blood transfusions), if anemia prompted a PCR test for B. gibsoni then it is little surprise that most were anemic. To get a true picture of the disease and the relative frequency of particular presentations, consecutive cases need to be collected to ensure a true representation of the various findings.

Response 1: Expanding over 10 years we initially obtained raw data for 264 dogs with a PCR diagnosis. Then due to missing data, age range selection…etc we excluded 156 dogs. We agree we only included dogs that had been diagnosed by PCR and for that reason this is not a representation of “all” dogs infected with B. gibsoni but includes “all” dogs that came to the clinic for any reason and were diagnosed by PCR and had the data we could analyze.

It is important that you provide this data to readers. You started with 264 and worked that down to 108. Explain how and why you did this in the M&M section.

It is important that you tell readers that the cases were not collected consecutively and as such you cannot draw any conclusions about how common the various observations are across all B gibsoni infected dogs are. The data is biased by the way cases were selected. Please say this in your discussion in the paragraph where you describe the weaknesses of the work.

I think it would be very helpful if in your introduction you made the statement early in the introduction that ‘because examination of a blood smear in an attempt to identify parasites is an insensitive means of diagnosing B. gibsoni infections (ref), it becomes important for clinicians to recognize a cluster of clinical, hematological and biochemical findings that are most consistent with this infection. In this study we provide evidence to support the use of a cluster of clinical and clinicopathological findings that can raise a clinician’s suspicion index for this infection and prompt early definitive PCR testing.’

All the data was collected on presentation, hence, before treatment. Most presented clinical signs like lethargy and anorexia, some had had ticks and that’s why they came and were tested. Surely the ones that presented with anemia and thrombocytopenia, fever, splenomegaly...etc would be suspicious of infection and tested. Dogs coming for routine visits or procedures that had the needed data were also included though and they would represent chronic or subclinical undiagnosed infections.

This is helpful information for the reader to have. Please put it in the M&M section.

Specific and supportive treatment depended on the case and included antiparasitic protocols and fluid therapy and blood products in cases where it was deemed necessary. In this study we aim to provide diagnostic findings information (rather than epidemiologic, therapeutic…) and add to the body of knowledge. We acknowledge there were many more dogs infected that were excluded or not diagnosed. The selection criteria aimed to standardize comparable data sets that would be more genuine, with less comorbidities or limited immune response due to age, as it would be much harder to interpret results otherwise. We agree the design of the study (being retrospective) is a limitation. We appreciate your comments and believe they will encourage us to conduct further focused studies. We have expanded the discussion to include these limitations and acknowledge the need of studying consecutive cases and of prospective designs.

I understand the purpose of the study more clearly now (because diagnosis is not straightforward in most cases, a cluster of findings is required to heighten suspicion and prompt PCR testing) and hence I understand why this data is not included (and needn't be).

Comments 2: There is no attempt to help the reader understand from the data what would make a case severe (complicated). What proportion of the cases described were severe (complicated) – and what made them severe (complicated). What proportion were mild (uncomplicated). There is no indication of outcome. What proportion of these cases survived? Required advanced hospital care?

Response 2: Thank you for your comment. We focused the aim of the study mostly in describing the most common clinical findings and the diagnostic findings. We did not collect subsequent data. There were 17 dogs with more severe anemia and TCP and 28 with marked anemia and moderate TCP. These two groups were the dogs that showed worse clinical signs. We have not investigated however whether there were fatalities. Out of the 108, none died on the day of presentation. Our aim is to guide clinicians towards the diagnosis rather than the classification of complicated. We surely had some more complicated unusual cases I can even remember from treating myself but many of them as it occurs in B. gibsoni were milder and would respond to therapy. I added some info in the discussion.

Agreed, thank you.

Comments 3: It is said that the most severe form of the disease is seen in young puppies (line 48) and yet this age group was excluded. Age plays an important role in the form of disease that presents and it seems odd that only cases 1 – 9 years of age were included. In other canine babesia infections, age plays a role in the form of disease seen.

Thank you

Response 3: We know that age is a risk factor and in a large study we found infections to be less common in dogs <10 years old (Muguiro et al. 2023). Additionally, we also know that although it’s true young puppies suffer a hyperacute or more severe presentation, it is also true this is a rare occurrence (Boozer et al 2005). We aimed to present the data of the most commonly affected age range, that would have the least interference with either chronic disease/comorbidities in older patients, or with the lack of maturity of; immune system, renal and hepatic function, blood pressure...etc in younger puppies making them in first place more likely to suffer complicated forms of this or any other disease. This would make a very interesting study focusing on young puppies alone.

Looking at the raw data the amount of comorbidities in older patients was remarkable and that’d have made the interpretation rather difficult.

You mention comorbidities in the older group of animals being the reason this age group was excluded. IT would be helpful if you provided a brief list (in the text) of examples of the co-morbidities you were looking to exclude. Cancer? Endocrinopathies? CKD? CHF?

Comments 4: What was done to try to ensure that dogs with B. gibsoni only had this single tick-borne pathogen? Co-infections with other parasites are not uncommon and do influence the clinical picture seen.

Response 4: Our PCR tests are all performed at CityU VDL (since July 2018) and previously at Pathlab diagnostics. Their tick fever panels include Babesia spp., B. gibsoni, B. canis and Ehrlichia canis. Before submitting the PCR, most will have an in-house test (Snap Iddexx 4DX including HW, EC, AP and Lyme). However, it is true not all cases might have had the full panels as being a retrospective study the full reports are not always attached. It is common practice; we could not even run B. gibsoni alone until very recently. We have included this information in the limitations.

There were 2 anaplasma +ve and 4 EC positive on serology. One had all 4DX positive results which were false positives for HW and for EC afterwards. All of them had mild or normal HCT and the disease was mild in presentation too. The diagnosis of babesiosis was always by a PCR panel as included in the manuscript.

Thank you.

Comments 5: There are several places where the English language needs an edit.

Response 5: We have edited extensively the manuscript to improve the language.

I will make a short list of some additional changes that I think could be made (below).

Comments 6: Line 42: Is there a ref for B. gibsoni being found locally transmitted in Africa? I am not – although imported cases have been reported it is not an indigenous disease in Africa as far as I am aware.

Response 6: Prof Schoeman has reported it as endemic in North and East Africa (Schoeman 2009).

The Schoeman paper is a review. And that paper references a review! You will not find a reference to autochthonous transmission of B. gibsoni in Africa in the literature as far as I know. The only references you will find will be to imported cases in Africa.

Comments 7: Line 48: Ref 12 does not refer to B. gibsoni. Although the reference calls the parasite B. canis, it is clear now that in fact it was B. rossi – which results in a much more severe disease more commonly than what B. gibsoni does.

Response 7: I updated the reference (Birkenheuer et al 2005) that reported fighting dogs suffered from this, the sentence has been rephrased. We have occasionally seen this type of presentations here too with DIC and SIRS published as a case report and there are other isolated reports too that I have added. 

Thank you

Comments 8: Line 57-58: Ref 2 is a poor ref to use here. You need to go back to an original (not a review) report that describes obvious organ dysfunction in B. gibsoni (I’d be interested in knowing if you could find one!).

Response 8: I agree with you that B. gibsoni is less commonly associated with organ dysfunction or at least in not widely reported as it happens with B. rossi. The sentence has been rephrased, and the references have been changed. There are several reports, early back to Conrad in 1991 to more recently in 2021 with complications including DIC, glomerulonephropathy, renal failure…etc. We have rephrased to reflect that rather than organ failure.

Thank you.

Comments 9: Line 60 – 61: Ref 17 again describes the disease caused by B. rossi (quite a different disease caused by B. gibsoni). Ref 12 is also a review – it is always best to refer to original work and not reviews for statements like this.

Response 9: The sentence has been rephrased and the references updated as per comment 8.  

Thank you.

Comments 10: It would be great if you could give an indication of the sensitivity of a peripheral blood smear vs. a PCR for the diagnosis. Why is there relatively little reliance on a blood smear for this infection when for other species of this same genus, a blood smear is the go-to diagnostic test.

Response 10: PCR has been reported to be very sensitivite, detecting DNA in 2.5 ul of blood with parasitemia of 0.000002% (Fukumoto 2001). However, the presence of piroplasm inside red blood cells by and experienced veterinarian or pathologist together with supporting signs would be a reliable diagnosis. We would always confirm with PCR though. B. gibsoni is very small and on many occasions, there are artifacts that can make its diagnosis difficult, especially when done in-house. Additionally, there are also other hemoparasites that could be mistakenly misdiagnosed; hence confirmation is by PCR, far more sensitive and specific. Additionally, in a retrospective study like ours with the analytics performed mostly in house there might be a large variability of operator skills when interpreting cytology if this was the only diagnostic method. 

I get this now – and that is why I think adding the statement I suggested above early in your intro would help readers understand why it is important for the cluster of findings to be described to heighten diagnostic suspicion (precisely because you could make a blood smear and not see parasites despite the dog having B gibsoni infection)

Comments 11: Is a slide review not standard practice when a complete blood count is performed? Surely in an area where there is a significant case load of a tick-borne pathogen, a slide review looking for the parasite should be standard practice? This would also be true to evaluate red cell morphology. How was the spherocyte percentage determined?

Response 11: We agree it would be ideal to run a blood smear after every CBC but unfortunately that is not the case in practice due to time restrains in many cases. This is another limitation that we mentioned due to the study design. Hence, the lower number of dogs that had a blood smear in the study. The smears were reviewed either by the clinicians in-house (like when the dog presented anemic to the ER) or by pathologists when sent to a referral lab for hematology.    

Thank you.

Comments 12: Paragraph beginning with line 102: the various cells counted in the CBC are listed but reticulocytes are not noted. How was the reticulocyte count established?

Response 12: Sorry. We had the reticulocyte count too counted by the hematology analyzer and is in the same paragraph following MCHC. However, we have deleted that paragraph following another reviewer comments as appeared redundant and in the paragraph above we mention we had CBC results and all of them have been now added to the table captions and appeared already in the results section.

Thank you.

Comments 13: Line 107: ‘citrate prothrombin time’ should read ‘prothrombin time on citrated blood (same for the PTT).

Response 13: Thank you for pointing this out. “APTT activated partial thromboplastin time on citrated blood, PT prothrombin time on citrated blood” was added in the caption of Table 5.

Thank you.

Comments 14: The method(s) used for the urinalysis are not made clear in this section of the M&M. Was there no evidence of hemoglobinuria? It would seem to me that this is a disease characterized by extravascular hemolysis more than intravascular hemolysis? Comment on this please as other canine Babesia parasites cause very obvious intravascular hemolysis and hemoglobinuria.

Response 14: Hemoglobinuria was not common and was described in the clinical notes meaning red/dark urine. This was actually pigmenturia that would include hematuria, hemoglobinuria and biliribinuria, which was actually the most common finding on urine tests. This would support extravascular hemolysis being more common. The sediment exam of urine was not reported in most cases hence we cannot report whether there were red cells in urine.  

Thank you.

Comments 15: Is ref 22 a dog-specific reference?

Response 15: Yes, it was. We have deleted and changed for a canine reference. Cornell eClinpath actually classifies TCP either subjectively or using human guidelines. Hence, we defined the ranges based on what are normal ranges in veterinary literature.

Thank you.

Comments 16: In the section on Statistical Analysis, no mention is made of whether the data was tested for normality – and yet means and SD’s are reported.

Response 16: We did test for normality (by skewness) but some of them have normal distribution while some do not. Means and SDs should only be reported in those with normal distribution.

Please make this clear in the text where you describe the statistical methods.

Comments 17: Line124: do you have any data on if the dogs were sterilized or not?

Response 17: Thank you for pointing this out. We have the relevant information, which is added

Thank you. Just for interest sake, there is an interesting study out now that shows the importance of this: Knobel DL, Hanekom J, van den Bergh MC, Leisewitz AL. Effects of gonadectomy on the incidence rate of babesiosis and the risk of severe babesiosis in dogs aged 6 months and older at a veterinary academic hospital in South Africa: A case-control and retrospective cohort study. Preventive Veterinary Medicine. 2024 Sep 1;230:106293.

Comments 18: Line 125: reporting the number of dogs of a certain breed are affected is not helpful if we have no idea about how common that breed is in the general population. If most dogs in the general population are mongrels then it is probably not surprising that most affected dogs are mongrels. I do not believe figure 1 is useful (it should be removed).

Response 18: Thank you for pointing this out. Figure 1 and text reporting breeds are removed.

Thank you.

Comments 19: Table 1: It is very interesting that fever was not a more common presenting clinical sign. Fever is a hallmark of a host cytokine response. This is something that should be discussed in the Discussion as it probably relates to why the disease caused by B. gibsoni is less severe than that caused by B rossi and B canis. I found it interesting that there was so little comparison made between the disease caused by B. gibsoni and that caused by other canine Babesia parasites.

Response 19: We have included this in the introduction and the discussion comparing with other babesias and adding references.

Thank you.

Comments 20: Table 2 and related text: it is not necessary to report on HCT, RBCC and HGB – they all tell us basically the same thing. Typically a clinician is most interested in the HCT.

Response 20: Thank you for pointing this out. Rows of data and text reporting RBC and HGB are deleted.

Thank you.

Comments 21: Table 4 should be removed. The text discussion is adequate.

Response 21: We appreciate we have a number of tables but we still believe it is a very easy and graphical way to see the correlation of HCT and PLT in this disease that is rather unique. In the sense that we hardly encounter a severely anemic dog with normal PLT count in B. gibsoni infections. To the point we would seriously consider other differentials rather than B. gibsoni as the cause of anemia.

I think you should consider do a correlation statistic between HCT and PLT to show this correlation statistically (such as a Spearman’s)

I still think having 2 tables that speak to the relationship between HCT and PLT is too much. I would still suggest removing Table 3, leaving table 4 and adding a correlation statistic to the text (and the test to the paragraph describing ‘Statistics’).

Comments 22: Paragraph starting with line 161: there is no indication of if there was an increase in band cells (left shift).

We did not had that parameter in the in-house analyzer but we acknowledge that monocytosis can be a common iatrogenic finding when using automatic analyzers when these could be in fact band neutrophils. We see that often in dogs receiving chemotherapy that appear neutropenic when in fact the bone marrow rebounded and are neutrophilic with false monocytosis recorded by the automatic measurement. This would need to be differentiated in further research but monocytosis has been reported in other similar studies assessing B. gibsoni clinicopathological features (possibly for the same reason) and is indeed what general practitioners will see in their analyzers even if iatrogenic. We suggest monocytosis might be falsely elevated when actually are band neutrophils. We included this in the manuscript.

Thank you.

Comments 23: In other hemolytic diseases (such as IMHA and the disease caused by B. rossi), there is a disproportionate increase of urea to creatinine. Was this true here? 

We had not calculated but we did after your comment and the results showed that 52% had BUN:CREA elevated. We have reflected that in the manuscript.

You could consider looking at the following references:

Piek CJ, Junius G, Dekker A, Schrauwen E, Slappendel RJ, Teske E. Idiopathic immunemediated hemolytic anemia: treatment outcome and prognostic factors in 149 dogs. Journal of veterinary internal medicine. 2008 Mar;22(2):366-73.

De Scally MP, Leisewitz AL, Lobetti RG, Thompson PN. The elevated serum urea: creatinine ratio in canine babesiosis in South Africa is not of renal origin. Journal of the South African Veterinary Association. 2006 Dec 1;77(4):175-8.

Comments 24: what I an intraerythrocytic inclusion? A parasite? A Heinz body? Nuclear remnant?

Response 24: Thank you for pointing this out. “Intraerythrocytic inclusion” is modified to “intraerythrocytic piroplasms”

Thank you.

Comments 25: Is there any sense of how long these dogs had been ill before being presented for care? The point is made that there may not have been enough time for the marrow to cause signs of red cell regeneration. It has been shown that in B. rossi the anemia (despite being obviously hemolytic) is inappropriately regenerative.

Response 25: The majority had normocytosis, normochromasia but reticulocytosis. In general the blood test was taken at presentation hence when they first came to the clinic. Since that would be early in the onset of the disease in most cases it is feasible more regenerative features were not present yet. 

Thank you.

Comments 26: References: There are numerous references where Babesia gibsoni should read Babesia gibsoni or where scientific names are not in italics.

Response 26: Thank you for pointing this out. Scientific names are modified according to binomial nomenclature.

Thank you

A few additional comments:

Line 18 (Abstract): ‘most common arthropod-borne infection…

Line 302: There were the 29 breeds … delete ‘the’.

Table 4: Please make clear what number is in the bracket. Is it the number of animals? And if so, out of how many?

Line 438: A ratio does not have units, it is simply a number.

Line 454: I think you should say “… describe the clinical and clinicopathological…’ – because that is what you did. You did not only look at lab values, you included valuable clinical observations.

Line 482: not only were very few dogs jaundiced, the total bili was also very seldom raised – which is quite different to other Babesia species infections and I think argues for a far slower rate of red cell turn over than with other Babesias.

Line 498: Why not include reticulocyte count in your list of regenerative features?

Paragraph beginning line 520: The most extensive work on the coagulopathy in Babesia infections has been done by Goddard and I feel you would be remiss by not referencing it:

Goddard A, Leisewitz AL, Kristensen AT, Schoeman JP. Platelet indices in dogs with Babesia rossi infection. Veterinary Clinical Pathology. 2015 Dec;44(4):493-7.

Goddard A, Wiinberg B, Schoeman JP, Kristensen AT, Kjelgaard-Hansen M. Mortality in virulent canine babesiosis is associated with a consumptive coagulopathy. The Veterinary Journal. 2013 May 1;196(2):213-7.

Line 532: Change ‘pre-renal’ to ‘non-renal’

Line 561: ‘… providing limited guidance in suspecting’ should be changed to ‘providing little help is supporting a suspicion of B. gibsoni infection’.

Line 573: Add ‘clinical and clinicopathological…’

Line 602: Add ‘..in a cohort of selected dogs infected with…’

Line 604: ‘diagnosis and treatment’. Remove ‘treatment’. You are not examining treatment in your work – you are examining diagnosis.

Line 631: ‘..patterns in clinical, hematological..’

Comments on the Quality of English Language

I have made a few suggestions in my review. There are a few places in the manuscript where the language use is not ideal but is does not obscure meaning. 

Author Response

Review #2: Title: Clinicopathological features of 108 dogs infected with Babesia gibsoni
Journal: Animals

B gibsoni case series review

I think this is a helpful piece of work and adds valuable data to the canine Babesia literature. The authors are to be encouraged for the investigation, and I would encourage them to carefully consider my further comments below.

I think the title could change to “Clinical and clinicopathological features of 108 dogs….”

We changed the tittle

Comments 1: The study was retrospective and included cases over a 10 year span. There is no indication of whether the cases were consecutive or not, and hence it seems very likely that there is a bias in the kind of cases that were included in the study, which will very likely be reflected in a bias in the data. For example, most cases were anemic but if all cases that presented with the diagnosis of B. gibsoni, and not just a random selection of cases that met the current inclusion criteria, would anemia and the spread of the degree of anemia have remained the same? Were the cases selected sick enough to require hospitalization? Were they all seen on an outpatient basis? Was the data collected before or after treatment (and what treatment, including supportive treatments like blood transfusions), if anemia prompted a PCR test for B. gibsoni then it is little surprise that most were anemic. To get a true picture of the disease and the relative frequency of particular presentations, consecutive cases need to be collected to ensure a true representation of the various findings.

Response 1: Expanding over 10 years we initially obtained raw data for 264 dogs with a PCR diagnosis. Then due to missing data, age range selection…etc we excluded 156 dogs. We agree we only included dogs that had been diagnosed by PCR and for that reason this is not a representation of “all” dogs infected with B. gibsoni but includes “all” dogs that came to the clinic for any reason and were diagnosed by PCR and had the data we could analyze.

It is important that you provide this data to readers. You started with 264 and worked that down to 108. Explain how and why you did this in the M&M section.

We included this in the M&M

It is important that you tell readers that the cases were not collected consecutively and as such you cannot draw any conclusions about how common the various observations are across all B gibsoni infected dogs are. The data is biased by the way cases were selected. Please say this in your discussion in the paragraph where you describe the weaknesses of the work.

We have added this limitation and agree there is a bias component. This seemed logical to us, in the sense we cannot interpret data that is missing and that might be affected by age and comorbidities. Therefore we agree with you we cannot draw firm conclusions as we don’t include “all” dogs infected with B. gibsoni.

I think it would be very helpful if in your introduction you made the statement early in the introduction that ‘because examination of a blood smear in an attempt to identify parasites is an insensitive means of diagnosing B. gibsoni infections (ref), it becomes important for clinicians to recognize a cluster of clinical, hematological and biochemical findings that are most consistent with this infection. In this study we provide evidence to support the use of a cluster of clinical and clinicopathological findings that can raise a clinician’s suspicion index for this infection and prompt early definitive PCR testing.’

We added this early in the discussion, thank you

All the data was collected on presentation, hence, before treatment. Most presented clinical signs like lethargy and anorexia, some had had ticks and that’s why they came and were tested. Surely the ones that presented with anemia and thrombocytopenia, fever, splenomegaly...etc would be suspicious of infection and tested. Dogs coming for routine visits or procedures that had the needed data were also included though and they would represent chronic or subclinical undiagnosed infections.

This is helpful information for the reader to have. Please put it in the M&M section.

Included now in the M&M data collection and results sections, thank you

Specific and supportive treatment depended on the case and included antiparasitic protocols and fluid therapy and blood products in cases where it was deemed necessary. In this study we aim to provide diagnostic findings information (rather than epidemiologic, therapeutic…) and add to the body of knowledge. We acknowledge there were many more dogs infected that were excluded or not diagnosed. The selection criteria aimed to standardize comparable data sets that would be more genuine, with less comorbidities or limited immune response due to age, as it would be much harder to interpret results otherwise. We agree the design of the study (being retrospective) is a limitation. We appreciate your comments and believe they will encourage us to conduct further focused studies. We have expanded the discussion to include these limitations and acknowledge the need of studying consecutive cases and of prospective designs.

I understand the purpose of the study more clearly now (because diagnosis is not straightforward in most cases, a cluster of findings is required to heighten suspicion and prompt PCR testing) and hence I understand why this data is not included (and needn't be).

Comments 2: There is no attempt to help the reader understand from the data what would make a case severe (complicated). What proportion of the cases described were severe (complicated) – and what made them severe (complicated). What proportion were mild (uncomplicated). There is no indication of outcome. What proportion of these cases survived? Required advanced hospital care?

Response 2: Thank you for your comment. We focused the aim of the study mostly in describing the most common clinical findings and the diagnostic findings. We did not collect subsequent data. There were 17 dogs with more severe anemia and TCP and 28 with marked anemia and moderate TCP. These two groups were the dogs that showed worse clinical signs. We have not investigated however whether there were fatalities. Out of the 108, none died on the day of presentation. Our aim is to guide clinicians towards the diagnosis rather than the classification of complicated. We surely had some more complicated unusual cases I can even remember from treating myself but many of them as it occurs in B. gibsoni were milder and would respond to therapy. I added some info in the discussion.

Agreed, thank you.

Comments 3: It is said that the most severe form of the disease is seen in young puppies (line 48) and yet this age group was excluded. Age plays an important role in the form of disease that presents and it seems odd that only cases 1 – 9 years of age were included. In other canine babesia infections, age plays a role in the form of disease seen.

Thank you

Response 3: We know that age is a risk factor and in a large study we found infections to be less common in dogs <10 years old (Muguiro et al. 2023). Additionally, we also know that although it’s true young puppies suffer a hyperacute or more severe presentation, it is also true this is a rare occurrence (Boozer et al 2005). We aimed to present the data of the most commonly affected age range, that would have the least interference with either chronic disease/comorbidities in older patients, or with the lack of maturity of; immune system, renal and hepatic function, blood pressure...etc in younger puppies making them in first place more likely to suffer complicated forms of this or any other disease. This would make a very interesting study focusing on young puppies alone.

Looking at the raw data the amount of comorbidities in older patients was remarkable and that’d have made the interpretation rather difficult.

You mention comorbidities in the older group of animals being the reason this age group was excluded. IT would be helpful if you provided a brief list (in the text) of examples of the co-morbidities you were looking to exclude. Cancer? Endocrinopathies? CKD? CHF?

We have included this in the discussion following the discussion of why we excluded older and younger dogs

Comments 4: What was done to try to ensure that dogs with B. gibsoni only had this single tick-borne pathogen? Co-infections with other parasites are not uncommon and do influence the clinical picture seen.

Response 4: Our PCR tests are all performed at CityU VDL (since July 2018) and previously at Pathlab diagnostics. Their tick fever panels include Babesia spp., B. gibsoni, B. canis and Ehrlichia canis. Before submitting the PCR, most will have an in-house test (Snap Iddexx 4DX including HW, EC, AP and Lyme). However, it is true not all cases might have had the full panels as being a retrospective study the full reports are not always attached. It is common practice; we could not even run B. gibsoni alone until very recently. We have included this information in the limitations.

There were 2 anaplasma +ve and 4 EC positive on serology. One had all 4DX positive results which were false positives for HW and for EC afterwards. All of them had mild or normal HCT and the disease was mild in presentation too. The diagnosis of babesiosis was always by a PCR panel as included in the manuscript.

Thank you.

Comments 5: There are several places where the English language needs an edit.

Response 5: We have edited extensively the manuscript to improve the language.

I will make a short list of some additional changes that I think could be made (below).

Comments 6: Line 42: Is there a ref for B. gibsoni being found locally transmitted in Africa? I am not – although imported cases have been reported it is not an indigenous disease in Africa as far as I am aware.

Response 6: Prof Schoeman has reported it as endemic in North and East Africa (Schoeman 2009).

The Schoeman paper is a review. And that paper references a review! You will not find a reference to autochthonous transmission of B. gibsoni in Africa in the literature as far as I know. The only references you will find will be to imported cases in Africa.

 Understood. We will only mention it has been detected but not that is endemic. Added an imported case reference.

Comments 7: Line 48: Ref 12 does not refer to B. gibsoni. Although the reference calls the parasite B. canis, it is clear now that in fact it was B. rossi – which results in a much more severe disease more commonly than what B. gibsoni does.

Response 7: I updated the reference (Birkenheuer et al 2005) that reported fighting dogs suffered from this, the sentence has been rephrased. We have occasionally seen this type of presentations here too with DIC and SIRS published as a case report and there are other isolated reports too that I have added. 

Thank you

Comments 8: Line 57-58: Ref 2 is a poor ref to use here. You need to go back to an original (not a review) report that describes obvious organ dysfunction in B. gibsoni (I’d be interested in knowing if you could find one!).

Response 8: I agree with you that B. gibsoni is less commonly associated with organ dysfunction or at least in not widely reported as it happens with B. rossi. The sentence has been rephrased, and the references have been changed. There are several reports, early back to Conrad in 1991 to more recently in 2021 with complications including DIC, glomerulonephropathy, renal failure…etc. We have rephrased to reflect that rather than organ failure.

Thank you.

Comments 9: Line 60 – 61: Ref 17 again describes the disease caused by B. rossi (quite a different disease caused by B. gibsoni). Ref 12 is also a review – it is always best to refer to original work and not reviews for statements like this.

Response 9: The sentence has been rephrased and the references updated as per comment 8.  

Thank you.

Comments 10: It would be great if you could give an indication of the sensitivity of a peripheral blood smear vs. a PCR for the diagnosis. Why is there relatively little reliance on a blood smear for this infection when for other species of this same genus, a blood smear is the go-to diagnostic test.

Response 10: PCR has been reported to be very sensitivite, detecting DNA in 2.5 ul of blood with parasitemia of 0.000002% (Fukumoto 2001). However, the presence of piroplasm inside red blood cells by and experienced veterinarian or pathologist together with supporting signs would be a reliable diagnosis. We would always confirm with PCR though. B. gibsoni is very small and on many occasions, there are artifacts that can make its diagnosis difficult, especially when done in-house. Additionally, there are also other hemoparasites that could be mistakenly misdiagnosed; hence confirmation is by PCR, far more sensitive and specific. Additionally, in a retrospective study like ours with the analytics performed mostly in house there might be a large variability of operator skills when interpreting cytology if this was the only diagnostic method. 

I get this now – and that is why I think adding the statement I suggested above early in your intro would help readers understand why it is important for the cluster of findings to be described to heighten diagnostic suspicion (precisely because you could make a blood smear and not see parasites despite the dog having B gibsoni infection)

We have added this early in the discussion

Comments 11: Is a slide review not standard practice when a complete blood count is performed? Surely in an area where there is a significant case load of a tick-borne pathogen, a slide review looking for the parasite should be standard practice? This would also be true to evaluate red cell morphology. How was the spherocyte percentage determined?

Response 11: We agree it would be ideal to run a blood smear after every CBC but unfortunately that is not the case in practice due to time restrains in many cases. This is another limitation that we mentioned due to the study design. Hence, the lower number of dogs that had a blood smear in the study. The smears were reviewed either by the clinicians in-house (like when the dog presented anemic to the ER) or by pathologists when sent to a referral lab for hematology.    

Thank you.

Comments 12: Paragraph beginning with line 102: the various cells counted in the CBC are listed but reticulocytes are not noted. How was the reticulocyte count established?

Response 12: Sorry. We had the reticulocyte count too counted by the hematology analyzer and is in the same paragraph following MCHC. However, we have deleted that paragraph following another reviewer comments as appeared redundant and in the paragraph above we mention we had CBC results and all of them have been now added to the table captions and appeared already in the results section.

Thank you.

Comments 13: Line 107: ‘citrate prothrombin time’ should read ‘prothrombin time on citrated blood (same for the PTT).

Response 13: Thank you for pointing this out. “APTT activated partial thromboplastin time on citrated blood, PT prothrombin time on citrated blood” was added in the caption of Table 5.

Thank you.

Comments 14: The method(s) used for the urinalysis are not made clear in this section of the M&M. Was there no evidence of hemoglobinuria? It would seem to me that this is a disease characterized by extravascular hemolysis more than intravascular hemolysis? Comment on this please as other canine Babesia parasites cause very obvious intravascular hemolysis and hemoglobinuria.

Response 14: Hemoglobinuria was not common and was described in the clinical notes meaning red/dark urine. This was actually pigmenturia that would include hematuria, hemoglobinuria and biliribinuria, which was actually the most common finding on urine tests. This would support extravascular hemolysis being more common. The sediment exam of urine was not reported in most cases hence we cannot report whether there were red cells in urine.  

Thank you.

Comments 15: Is ref 22 a dog-specific reference?

Response 15: Yes, it was. We have deleted and changed for a canine reference. Cornell eClinpath actually classifies TCP either subjectively or using human guidelines. Hence, we defined the ranges based on what are normal ranges in veterinary literature.

Thank you.

Comments 16: In the section on Statistical Analysis, no mention is made of whether the data was tested for normality – and yet means and SD’s are reported.

Response 16: We did test for normality (by skewness) but some of them have normal distribution while some do not.  Means and SDs should only be reported in those with normal distribution.

Please make this clear in the text where you describe the statistical methods.

Calculated Q2/median to show those that had not normal distribution and has been added

Comments 17: Line124: do you have any data on if the dogs were sterilized or not?

Response 17: Thank you for pointing this out. We have the relevant information, which is added

Thank you. Just for interest sake, there is an interesting study out now that shows the importance of this: Knobel DL, Hanekom J, van den Bergh MC, Leisewitz AL. Effects of gonadectomy on the incidence rate of babesiosis and the risk of severe babesiosis in dogs aged 6 months and older at a veterinary academic hospital in South Africa: A case-control and retrospective cohort study. Preventive Veterinary Medicine. 2024 Sep 1;230:106293.

Thank you for this info. Very interesting. Both males and females. We have very few entire male and females in the setting we work but we definitely need to look into that in rural areas around here. 

Comments 18: Line 125: reporting the number of dogs of a certain breed are affected is not helpful if we have no idea about how common that breed is in the general population. If most dogs in the general population are mongrels then it is probably not surprising that most affected dogs are mongrels. I do not believe figure 1 is useful (it should be removed).

Response 18: Thank you for pointing this out. Figure 1 and text reporting breeds are removed.

Thank you.

Comments 19: Table 1: It is very interesting that fever was not a more common presenting clinical sign. Fever is a hallmark of a host cytokine response. This is something that should be discussed in the Discussion as it probably relates to why the disease caused by B. gibsoni is less severe than that caused by B rossi and B canis. I found it interesting that there was so little comparison made between the disease caused by B. gibsoni and that caused by other canine Babesia parasites.

Response 19: We have included this in the introduction and the discussion comparing with other babesias and adding references.

Thank you.

Comments 20: Table 2 and related text: it is not necessary to report on HCT, RBCC and HGB – they all tell us basically the same thing. Typically a clinician is most interested in the HCT.

Response 20: Thank you for pointing this out. Rows of data and text reporting RBC and HGB are deleted.

Thank you.

Comments 21: Table 4 should be removed. The text discussion is adequate.

Response 21: We appreciate we have a number of tables but we still believe it is a very easy and graphical way to see the correlation of HCT and PLT in this disease that is rather unique. In the sense that we hardly encounter a severely anemic dog with normal PLT count in B. gibsoni infections. To the point we would seriously consider other differentials rather than B. gibsoni as the cause of anemia.

I think you should consider do a correlation statistic between HCT and PLT to show this correlation statistically (such as a Spearman’s)

I still think having 2 tables that speak to the relationship between HCT and PLT is too much. I would still suggest removing Table 3, leaving table 4 and adding a correlation statistic to the text (and the test to the paragraph describing ‘Statistics’).

We removed table 3. We understand. We believe as per your original comment one table is sufficient and is explained in the text already.

Comments 22: Paragraph starting with line 161: there is no indication of if there was an increase in band cells (left shift).

We did not had that parameter in the in-house analyzer but we acknowledge that monocytosis can be a common iatrogenic finding when using automatic analyzers when these could be in fact band neutrophils. We see that often in dogs receiving chemotherapy that appear neutropenic when in fact the bone marrow rebounded and are neutrophilic with false monocytosis recorded by the automatic measurement. This would need to be differentiated in further research but monocytosis has been reported in other similar studies assessing B. gibsoni clinicopathological features (possibly for the same reason) and is indeed what general practitioners will see in their analyzers even if iatrogenic. We suggest monocytosis might be falsely elevated when actually are band neutrophils. We included this in the manuscript.

Thank you.

Comments 23: In other hemolytic diseases (such as IMHA and the disease caused by B. rossi), there is a disproportionate increase of urea to creatinine. Was this true here? 

We had not calculated but we did after your comment and the results showed that 52% had BUN:CREA elevated. We have reflected that in the manuscript.

You could consider looking at the following references:

Piek CJ, Junius G, Dekker A, Schrauwen E, Slappendel RJ, Teske E. Idiopathic immunemediated hemolytic anemia: treatment outcome and prognostic factors in 149 dogs. Journal of veterinary internal medicine. 2008 Mar;22(2):366-73.

De Scally MP, Leisewitz AL, Lobetti RG, Thompson PN. The elevated serum urea: creatinine ratio in canine babesiosis in South Africa is not of renal origin. Journal of the South African Veterinary Association. 2006 Dec 1;77(4):175-8.

Thank you. We had already included this reference when we looked into it as per your suggestion

Comments 24: what I an intraerythrocytic inclusion? A parasite? A Heinz body? Nuclear remnant?

Response 24: Thank you for pointing this out. “Intraerythrocytic inclusion” is modified to “intraerythrocytic piroplasms”

Thank you.

Comments 25: Is there any sense of how long these dogs had been ill before being presented for care? The point is made that there may not have been enough time for the marrow to cause signs of red cell regeneration. It has been shown that in B. rossi the anemia (despite being obviously hemolytic) is inappropriately regenerative.

Response 25: The majority had normocytosis, normochromasia but reticulocytosis. In general the blood test was taken at presentation hence when they first came to the clinic. Since that would be early in the onset of the disease in most cases it is feasible more regenerative features were not present yet. 

Thank you.

Comments 26: References: There are numerous references where Babesia gibsoni should read Babesia gibsoni or where scientific names are not in italics.

Response 26: Thank you for pointing this out. Scientific names are modified according to binomial nomenclature.

Thank you

A few additional comments:

Line 18 (Abstract): ‘most common arthropod-borne infection… Done, than you

Line 302: There were the 29 breeds … delete ‘the’. Done, thank you

Table 4: Please make clear what number is in the bracket. Is it the number of animals? And if so, out of how many? Done and explanation added to caption and title. Thank you

Line 438: A ratio does not have units, it is simply a number. Corrected, thank you

Line 454: I think you should say “… describe the clinical and clinicopathological…’ – because that is what you did. You did not only look at lab values, you included valuable clinical observations. Added, thank you

Line 482: not only were very few dogs jaundiced, the total bili was also very seldom raised – which is quite different to other Babesia species infections and I think argues for a far slower rate of red cell turn over than with other Babesias.

We agree and we have added a few words and a reference to reflect that

Line 498: Why not include reticulocyte count in your list of regenerative features?

We agree and included it too, thank you

Paragraph beginning line 520: The most extensive work on the coagulopathy in Babesia infections has been done by Goddard and I feel you would be remiss by not referencing it:

Goddard A, Leisewitz AL, Kristensen AT, Schoeman JP. Platelet indices in dogs with Babesia rossi infection. Veterinary Clinical Pathology. 2015 Dec;44(4):493-7.

Goddard A, Wiinberg B, Schoeman JP, Kristensen AT, Kjelgaard-Hansen M. Mortality in virulent canine babesiosis is associated with a consumptive coagulopathy. The Veterinary Journal. 2013 May 1;196(2):213-7.

We appreciate your comment and the work Goddard et al. have done and we have added the references.

Line 532: Change ‘pre-renal’ to ‘non-renal’

Done. Thank you

Line 561: ‘… providing limited guidance in suspecting’ should be changed to ‘providing little help is supporting a suspicion of B. gibsoni infection’.

Corrected. Thank you

Line 573: Add ‘clinical and clinicopathological…’ Done

Line 602: Add ‘..in a cohort of selected dogs infected with…’ Done. Thank you

Line 604: ‘diagnosis and treatment’. Remove ‘treatment’. You are not examining treatment in your work – you are examining diagnosis. That is correct, thank you

Line 631: ‘..patterns in clinical, hematological..’ Added, thank you

Comments on the Quality of English Language

I have made a few suggestions in my review. There are a few places in the manuscript where the language use is not ideal but is does not obscure meaning. 

We did another additional grammar review

Reviewer 3 Report

Comments and Suggestions for Authors

Dera authors, 

i have no additional comments/recommendation. The updated MS can be accepted in the present form. 

Author Response

Thank you for your review

We appreciate your input and comments.

They have helped to improve the manuscript